# Boosting organic phosphorescence in adaptive host-guest materials by hyperconjugation

Huili Ma[1], Lishun Fu[1], Xiaokang Yao[1,2], Xueyan Jiang[1], Kaiqi Lv[1], Qian Ma[1], Huifang Shi [1], Zhongfu An [1,2] ✉ & Wei Huang [1,2,3] ✉

Phosphorescence is ubiquitous in heavy atom-containing organic phosphors, which attracts considerable attention in optoelectronics and bioelectronics. However, heavy atom-free organic materials with efficient phosphorescence are rare under ambient conditions. Herein, we report a series of adaptive host-guest materials derived from dibenzo-heterocyclic analogues, showing host-dependent color-tunable phosphorescence with phosphorescence efficiency of up to 98.9%. The adaptive structural deformation of the guests arises from the hyperconjugation, namely the n→π* interaction, enabling them to inhabit the cavity of host crystals in synergy with steric effects. Consequently, a perfect conformation match between host and guest molecules facilitates the suppression of triplet exciton dissipation, thereby boosting the phosphorescence of these adaptive materials. Moreover, we extend this strategy to a ternary host-guest system, yielding both excitation- and time-dependent phosphorescence with a phosphorescence efficiency of 92.0%. This principle provides a concise way for obtaining efficient and color-tunable phosphorescence, making a major step toward potential applications in optoelectronics.

Organic phosphorescence has drawn great attention benefiting from the potential applications in organic light-emitting diodes, bio-imaging, anti-counterfeiting, and so on[1–6]. However, it is normally unobservable in purely organic materials because phosphorescence is a forbidden transition between states of different spin multiplicity[7]. Since the heavy-atom effect can make a fast intersystem crossing process, great effort has been devoted to developing heavy atom-containing organic materials (e.g., Br, I, etc.) for boosting room-temperature phosphorescence (RTP)[8–10]. For heavy atom-free organic materials, incorporation of n-electron groups was adopted to obtain RTP in crystalline state[11–16]. Whereas, it is less powerful than the heavy-atom effect[17]. Besides the crystallization of luminogens, host–guest doping is an alternative approach for improving RTP[18,19], which can efficiently prevent aggregation-caused quenching in crystal. However,

the mismatch of the size and shape of guests to the cavity of the hosts results in an inevitable nonradiative deactivation, which enormously limits the improvement of RTP efficiency (Supplementary Fig. 1).

For example, the heteroatom-bridged heterocycles involve (n, π*) character, which not only greatly accelerates the ISC process for populating triplet excitons[20], but also promotes the radiative transition for boosting phosphorescence emission[21]. However, the intense phosphorescence can be only observed under harsh conditions (e.g., low temperature and deoxygenated atmosphere)[22,23]. As yet, these phosphors still have low RTP efficiency, even in a rigid molecular environment[24]. On the other hand, the heterocyclic molecules with flexible skeletons have the potential to achieve host-dependent conformations (Fig. 1a), presenting a promising avenue to reduce the mismatch between guest and host.

[1]Key Laboratory of Flexible Electronics (KLoFE) & Institute of Advanced Materials (IAM), Nanjing Tech University, 30 South Puzhu Road, Nanjing 211816, China. [2]The Institute of Flexible Electronics (IFE, Future Technologies), Xiamen University, Xiamen, China. [3]Frontiers Science Center for Flexible Electronics (FSCFE), MIIT Key Laboratory of Flexible Electronics (KLoFE), Northwestern Polytechnical University, Xi'an 710072, China. ✉e-mail: iamzfan@njtech.edu.cn; vc@nwpu.edu.cn

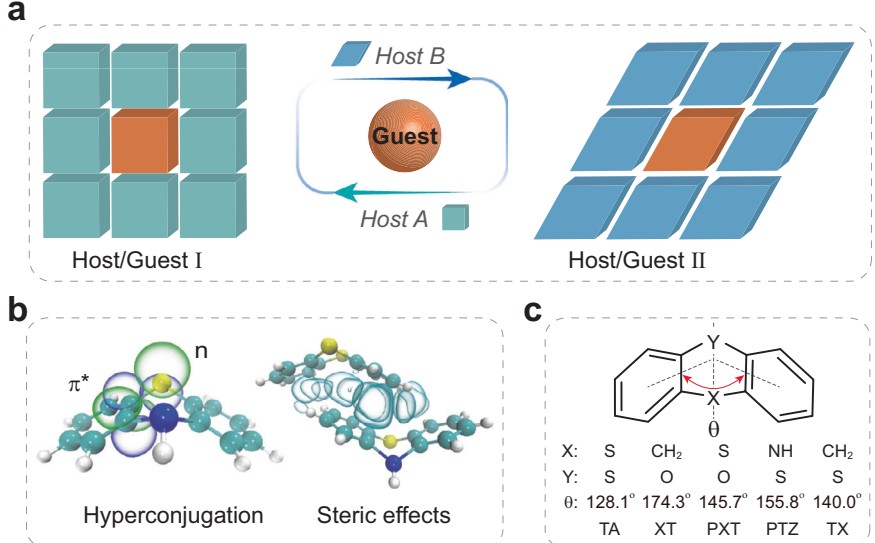

**Fig. 1 | Schematic illustration for the adaptive host–guest materials.**
**a** Demonstration of adaptation in host–guest systems. The guest molecule undergoes an adaptive conformation in response to the hosts with different conformational spaces. **b** Hyperconjugation and steric effects. Phenothiazine (PTZ) as prototype, note that manipulation of electron delocalization between the specific orbitals within a molecule and neighboring molecules could controllably tune the hyperconjugation and steric effects, respectively. The green and purple bubbles represent the n and $\pi^*$ wavefunctions with opposite phases, and the cyan bubbles are intermolecular interactions between host and guest molecules. **c** Molecular structures with flexible frameworks for thianthrene (TA), phenoxathiin (PXT), xanthene (XT), PTZ, and thioxanthene (TX). The dihedral angles (θ) are calculated between planes including atoms X and Y as well as the center of mass of benzene rings.

Notably, the n→$\pi^*$ interaction, one type of hyperconjugation (Supplementary Fig. 2), represents electron transfer from an occupied n to unoccupied $\pi^*$ orbitals, providing a significant contribution to stabilizing the heterocycles. Such hyperconjugation caused by electron delocalization between the specific orbitals can make a flexible bending/distorted structure[25,26]. When going to host–guest doped crystals, the steric effects arising from repulsive forces between overlapping electron clouds occur in π–π stacking of host and guest molecules (Fig. 1b), which is formed by the strong attractive interactions between adjacent aromatic rings[27]. This distorted electron distribution caused by steric effects will react on the hyperconjugation, thus resulting in a conformational adjustment to perfectly match the cavity of the host[28]. It will be beneficial to suppress nonradiative transition. Therefore, we reasoned that the adaptive host–guest materials might boost RTP efficiency significantly. Meanwhile, unexpected color-tunable RTP might occur owing to the conformation variation of phosphors.

To validate our hypothesis, here we prepared several adaptive host–guest materials using five typical prototype molecules, which consisted of heteroatoms and π-conjugation moiety (Fig. 1c). Thianthrene (TA), phenoxathiin (PXT), xanthene (XT), phenothiazine (PTZ), and thioxanthene (TX) are selected as host and guest molecules, which have flexible bending freedom along one of the principal molecular axes with different dihedral angles (θ, 128.1°, 145.7°, 174.3°, 155.8° and 140.0° for TA, PXT, XT, PTZ, and TX, respectively). By embedding PTZ guests into a crystalline host matrix, we obtained a series of adaptive host–guest materials with high RTP efficiency, reaching 96.8% for PTZ/PXT, along with a color-tunable RTP ranging from cyan, green to yellow-green. Such a principle was further evidenced by extended systems PXT/XT and PXT/TX, displaying a higher RTP efficiency of up to 98.9%. Moreover, this strategy succeeded in a ternary system (PTZ–PXT/TX), which shows green RTP with an efficiency of 92.0%. This strategy paves a feasible way to achieve highly efficient and color-tunable RTP in heavy atom-free organic materials for potential applications in organoelectronics and biomedicines via adaptive host–guest chemistry.

## Results
### Photophysical properties
First, the photophysical properties of five prototype molecules of TA, PXT, XT, TX, and PTZ were investigated in dilute solution and solid state. In dilute 2-methyltetrahydrofuran (2-mTHF) solution ($2 \times 10^{-5}$ M), all show two absorption peaks at around 250 and 300 nm (Supplementary Fig. 3), which are assigned as π–$\pi^*$ and n–$\pi^*$ characteristics, respectively. There only exists weak fluorescence with maximum peaks from 430 to 470 nm (Supplementary Figs. 3, 4 and Supplementary Table 1) under ambient condition. In contrast, at 77 K, a bright phosphorescence band appears at 400–500 nm with energy levels in ascending order: PTZ < TA < PXT < TX < XT (Supplementary Fig. 5). Impressively, the steady-state photoluminescence (PL) and phosphorescence spectra overlap almost completely (Supplementary Fig. 5), along with high PL efficiency (e.g., 96.1% for PXT and 78.0% for PTZ), indicating an extremely efficient ISC for populating triplet excitons, which is thus insensitive to the heavy-atom effect. However, the phosphorescence efficiencies of these phosphors are <4% in crystal (Supplementary Fig. 6 and Supplementary Tables 2–4).

We then probed the photophysical properties of the host–guest materials, named PTZ/TA, PTZ/PXT, and PTZ/XT, under the ambient condition as depicted in Fig. 2. Notably, PTZ is a guest, while TA, PXT, and XT are assigned as hosts. After optimizing guest doping concentrations (0.5 mol% for PTZ/TA, and 1.0 mol% for PTZ/PXT and PTZ/XT), we found that all host–guest phosphors showed strong PL bands at ca. 500 nm, along with faint fluorescence peaks at around 420 nm with lifetimes in the scope of ns-scale (Supplementary Figs. 7–10 and Supplementary Tables 2–5). Impressively, the emission colors of the host–guest materials are almost unchanged under excitation source on and off, but highly dependent on the crystalline host matrices, varying from cyan in PTZ/TA, green in PTZ/PXT to yellow-green in PTZ/XT (Fig. 2a, b). After a decay time of 8 ms, PTZ/TA shows a maximum peak at 498 nm with a lifetime of 29.74 ms, indicating RTP nature. For PTZ/PXT and PTZ/XT, the maximum RTP peaks are red-shifted to 508 nm with a lifetime of 44.81 ms and 517 nm with a lifetime of 41.44 ms, respectively. Figure 2d shows that the main

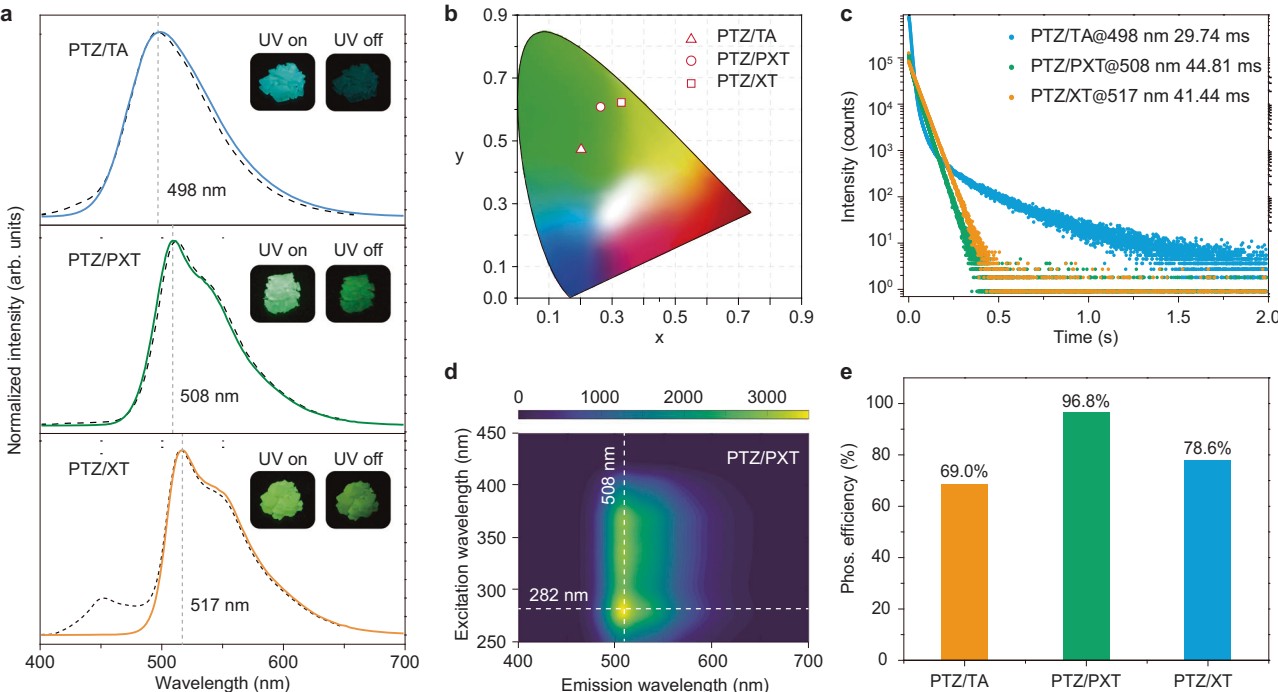

**Fig. 2 | Photophysical properties for the adaptive host–guest materials of PTZ/TA, PTZ/PXT, and PTZ/XT under ambient conditions. a** Steady-state photoluminescence (dash line) and phosphorescence (solid line) spectra with a decay time of 8 ms. Inserts: photographs of the phosphors under a 365 nm lamp on (left) and off (right). **b** CIE coordinate diagram of phosphorescence. **c** Lifetime decay profiles of emission bands at 498, 508, and 517 nm. The average lifetimes are also listed. **d** Excitation-phosphorescence emission mapping of PTZ/PXT. **e** Histogram of phosphorescence quantum yields. Doping concentrations were 0.5 mol% for PTZ/TA, 1.0 mol% for PTZ/PXT and PTZ/XT, respectively.

phosphorescence peak around 508 nm for PTZ/PXT remained unchanged by varying excitation from 250 to 450 nm, with the optimum excitation wavelength at 282 nm, implying the dopant PTZ should be uniformly dispersed in the PXT host. Notably, the profiles of phosphorescence spectra nearly overlap with PL spectra, especially for PTZ/PXT, indicating the ISC process is highly efficient. As expected, the phosphorescence efficiencies are as high as 69.0% for PTZ/TA, 96.8% for PTZ/PXT, and 78.6% for PTZ/XT, which are insensitive to the oxygen environment (Fig. 2e and Supplementary Figs. 11–14). To the best of our knowledge, this is the highest phosphorescence efficiency in purely organic phosphors under ambient conditions.

**Mechanism for efficient RTP emission**

To gain insight into the high efficiency and color tunability of RTP in the adaptive host–guest materials, we first conducted a series of control experiments on powder X-ray diffraction (XRD). It is easily found that the host–guest materials have similar diffraction peaks with those of the host crystals (Supplementary Fig. 15), indicating the guest doping has little influence on the crystalline morphology of the host materials. TA, PXT, and XT molecules as the hosts have diverse dihedral angles ($\theta$) of 128.1°, 145.7°, and 174.3° in the crystalline state, respectively, providing different cavities to accommodate the PTZ guest ($\theta_{PTZ} = 155.8°$). That is, the PTZ guest could adjust the conformation to match the host molecules appropriately. Herein the difference $\theta$ between host and guest molecules is defined as $\Delta\theta_{HG}$, representing the matching degree. The smaller $\Delta\theta_{HG}$ is, the better the conformation matches. The conformation match is increased from PTZ/TA, PTZ/XT to PTZ/PXT due to the $\Delta\theta_{HG}$ decreasing from 27.7°, 18.5° to 10.1°, which is consistent with the RTP efficiency rising from 69.0% for PTZ/TA, 78.6% for PTZ/XT to 96.8% for PTZ/PXT. Consequently, we reasoned that the RTP efficiency might be closely correlated to the conformation matching degree.

Subsequently, the inherent correlation between the structural deformation and the adaptive RTP behaviors was investigated with the

quantum mechanics/molecular mechanics (QM/MM) model (Supplementary Fig. 16). By evaluating the geometric and electronic structures of these host–guest systems at the level of (TD)B3LYP-D3(BJ)/def2-SVP, it is found that, from PTZ/TA, PTZ/PXT to PTZ/XT, the PTZ guest molecule shows a host-dependent dihedral angle $\theta_{PTZ}$, ranging from 137.8°, 147.2° to 177.8°. For the host molecules, there are almost no changes in dihedral angles (Supplementary Table 6). In other words, the guest PTZ undergoes a large structural deformation ($\Delta\theta_{PTZ}$) in response to the molecular environments of the host matrices due to the small conformational reorganization energies (<2.5 kcal/mol, see Supplementary Table 7), varying from 18.0°, 8.6° to 22.0°, compared to that of PTZ in single-crystal ($\theta_{PTZ} = 155.8°$). Moreover, this change also makes a better conformation match between host and guest molecules, which is beneficial to confine the molecular motions. Because $\Delta\theta_{HG}$ is reduced to 9.7° for PTZ/TA, 1.5° for PTZ/PXT, and 3.5° for PTZ/XT (Fig. 3a), respectively.

Such a structural deformation originated from the synergy of the hyperconjugation and steric effects (Fig. 3b–d, Supplementary Figs. 17, 18 and Supplementary Tables 8, 9). The hyperconjugation stabilizes a flexible bending structure for PTZ, which is determined by the delocalization of electrons from heteroatoms to nearby benzene rings with the energy of −115.26 kcal/mol, especially the n→π* interaction from nitrogen (−30.83 kcal/mol) and sulfur (−15.52 kcal/mol) atoms. When embedding PTZ into host crystals, the reduced density gradient[29] and energy decomposition analyses[29] demonstrated that the repulsive force ($E_{rep}$) makes the change in dihedral angles of $\theta_{PTZ}$ caused by the PTZ-host π–π stacking, which was mainly induced by the dispersion interaction ($E_{disp}$), along with a tiny electrostatic interaction ($E_{ele}$). Remarkably, from PTZ/TA, PTZ/PXT to PTZ/XT, the variation of these intermolecular interactions (e.g., $E_{rep}$ varies from 47.26, 39.65 to 59.80 kcal/mol) agrees well with the changes in structural deformation of $\Delta\theta_{PTZ}$ (from 18.0°, 8.6° to 21.9°). Moreover, the delocalization energies are decreased from −99.34 kcal/mol for PTZ/TA, −108.44 kcal/mol for PTZ/PXT to −126.36 kcal/mol for PTZ/XT,

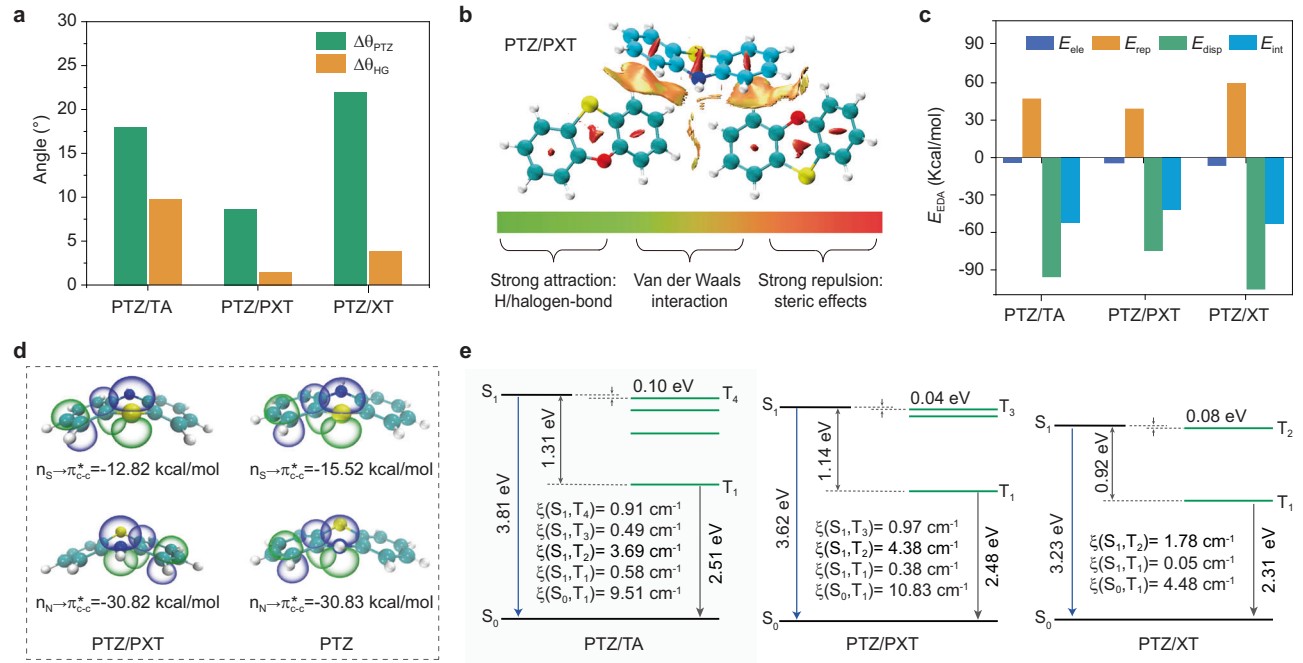

**Fig. 3 | Illustration of RTP mechanism for the adaptive host–guest materials. a** Difference of dihedral angles of PTZ ($\Delta\theta_{PTZ}$) in PTZ crystal and host–guest systems, as well as dihedral angles ($\Delta\theta_{HG}$) between the PTZ and host molecules. **b**, **c** The reduced density gradient (RDG) isosurface, and the energy decomposition analysis (EDA) for the intermolecular interaction ($E_{int}$) between the PTZ and host molecules. Notably, the $E_{int}$ can be divided into electrostatic ($E_{ele}$), repulsive ($E_{rep}$), and dispersion ($E_{disp}$) forces, respectively. **d** Vicinal π-type $n_X \rightarrow \pi^*_{C-C}$ (X = S/N), hyperconjugation interactions (and associated $E^{(2)}$ stabilization energy) in the PTZ/PXT and PTZ crystal. **e** Energy diagrams and spin–orbit coupling matrix elements (ξ) for the PTZ/TA, PTZ/PXT, and PTZ/XT.

indicating the reduction of electrons from heteroatoms to nearby benzene rings (Supplementary Table 8 and Supplementary Fig. 17), making a more planar and rigid structure, thus it is responsible for the increased dihedral angles of the PTZ guest ($\theta_{PTZ}$) in PTZ/TA, PTZ/PXT to PTZ/XT. Namely, the increased stabilization energy induced by structural deformation makes it difficult for the $T_1$ states of PTZ guest to undergo the structural relaxation, which was confirmed by the decrease of the reorganization energies ($\lambda$) from 0.575 eV in PTZ/TA, 0.487 eV in PTZ/PXT to 0.256 eV in PTZ/XT. Therefore, we concluded that the structural deformation of the guest in the doping crystals could be ascribed to the synergy of hyperconjugation and steric effects.

With the increase of the dihedral angles ($\theta_{PTZ}$) for the PTZ molecules from PTZ/TA, PTZ/PXT to PTZ/XT, Fig. 3e showed that (i) there are small energy gaps between $S_1$ and nearby $T_n$ states (from 0.10, 0.04 to 0.08 eV), and large SOC matrix elements of ξ($S_1$, $T_2$) with values from 3.69, 4.38 to 1.78 cm⁻¹, resulting in a fast ISC process for populating triplet excitons, which is favorable to the high phosphorescence efficiency; (ii) the energy levels of $T_1$ states are gradually decreased from 2.51 to 2.48 to 2.31 eV, which agrees well with the red-shift of RTP spectra in the experiment. Taken together, we proposed that the hyperconjugation caused by heteroatoms endowed emitter PTZ with flexible bending freedom. Then it underwent an adaptive structural deformation to accommodate the cavity of the crystalline host matrix in conjugation with the steric effects, thereby generating highly efficient and color-tunable RTP in these adaptive host–guest materials.

### Extension of the adaptive host–guest materials with RTP

To prove the generality of our strategy, we constructed two PXT-doping systems (1.0 mol%), i.e., PXT/TX and PXT/XT. In consideration of $T_1$ energy levels (PXT < TX < XT, Supplementary Fig. 5). By looking into the conformation match, we found the guest PXT (θ = 145.7°) is well matched by TX (θ = 140.0°), then XT (θ = 174.3°). As anticipated, the RTP color changes from blue to sky blue for PXT/TX and PXT/XT,

respectively (Fig. 4). The RTP bands are red-shifted from 457 nm with a lifetime of 47.91 ms to 480 nm with a lifetime of 22.48 ms (Fig. 4b). The RTP efficiency is decreased from 98.9% for PXT/TX to 54.9% for PXT/XT, this can be ascribed to the reduced conformation match $\Delta\theta_{HG}$, increasing from 5.7° to 28.6°. To explore the guest-dependent RTP property, we prepared PTZ/TX, which exhibits an RTP band around 505 nm and a lifetime of 39.5 ms. In contrast to PXT/TX, the RTP efficiency of PTZ/TX decreases to 84.3%, mainly attributed to the reduced conformation match ($\Delta\theta_{HG}$ = 15.8°). Based on the dibenzo-heterocyclic analogs, we strongly demonstrated that the conformation match in the adaptive host–gust materials governs the phosphorescence efficiency (Fig. 4e), which is mainly attributed to the nonradiative transitions (Supplementary Table 10).

Beyond two-component doping, we bring this strategy to a ternary host–guest system PT/TX, i.e., embedding PTZ and PXT guests into the crystalline TX matrix. Since the $T_1$ energy levels are in order of PTZ < PXT < TX (Supplementary Fig. 5), PXT and PTZ are chosen as guests, and TX is defined as a host. To avert the energy transfer from PXT to PTZ, the doping concentrations of PTZ and PXT are fixed as low as 0.001 and 0.01 mol%, respectively. Compared to the sky blue RTP in PXT/TX (456 nm), the ternary host–guest system displays green RTP after a decay time of 8 ms, owing to the integration of RTP peaks around 460 and 496 nm along with lifetimes of 17.73 and 45.91 ms, respectively (Fig. 4a and Supplementary Fig. 20). The former emission band is assigned to the PXT molecule, while the latter one is originated from the PTZ compound. Because it is in line with phosphorescence peaks of PXT and PTZ in a single molecular state at 77 K (Supplementary Fig. 5). Figure 4d showed that the strongest phosphorescence peaks are changed from 456 to 498 nm with excitations variation from 250 to 450 nm. Moreover, as decay time went on, the intensity of the RTP band at around 507 nm gradually became stronger than that of 460 nm (Supplementary Figs. 21, 22). These features demonstrated that this ternary system had dual phosphorescence emission, originating from the PTZ and PXT molecules, respectively. Benefitting from

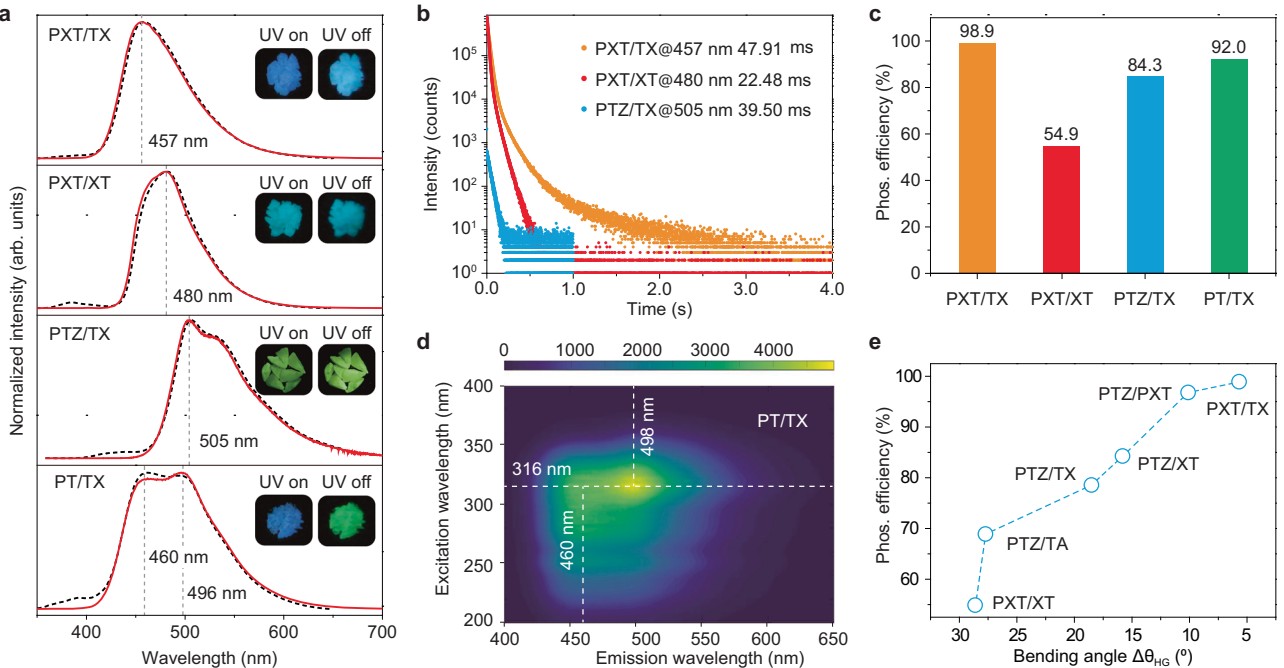

**Fig. 4 | Photophysical properties of the PXT/XT, PXT/TX, PTZ/TX, and PT/TX under ambient conditions. a** Steady-state photoluminescence (dash line) and phosphorescence (solid line) spectra. Inserts: photographs of the doping phosphors taken under a 365 nm lamp on and off. **b** Lifetime profiles of emission bands at 457, 480, and 505 nm. The average lifetimes are also listed. **c** Histogram of phosphorescence quantum yields. **d** Excitation-phosphorescence emission mapping of the PT/TX material. **e** Phosphorescence efficiency versus conformation match ($\Delta\theta_{HG}$). Note that both PTZ and PXT are guests for PT/TX.

the similar conformation match among PTZ ($\theta$ = 158.0°), PXT (147.7°), and TX (140.0°), we obtained a high RTP efficiency of 92.0%. To the best of our knowledge, this is the first ternary system with highly efficient RTP, demonstrating the broad expansibility of such a feasible approach. This strategy will pave an avenue to develop new organic RTP phosphors with multi-functions[30].

## Discussion

In summary, we have proposed a feasible approach to develop adaptive host–guest materials for achieving highly efficient and color-tunable RTP. The flexible frameworks of organic analogs caused by heteroatoms are essential to such a principle, which not only enables the guest to take an adaptive structural deformation for the accommodation of the host cavity but also accelerates the ISC process for populating triplet excitons. With optimization of the doping concentrations, the adaptive host–guest material of PXT/TX shows the highest RTP efficiency of up to 98.9%. As the host molecules changed, color-tunable RTP was observed owing to the adaptivity of guests to the hosts, which was attributed to the synergy of hyperconjugation and steric effects. Moreover, we succeeded in bringing this strategy to a ternary host–guest system, displaying both excitation- and time-dependent RTP with a high efficiency of 92.0%. This work not only provides a concise principle for promoting RTP efficiency and manipulating RTP color but also paves a new channel to design organic RTP materials toward adaptivity and multi-function for optoelectronic applications.

## Methods

### Reagents and materials

All chemical reagents were purchased from Energy Chemical and were purified by column chromatography followed by recrystallization three times. The purity was verified by HPLC. All purified guest and host molecules were (molguest:molhost = 5%:95%) ground and mixed in a mortar, then added to a 20 ml quartz tube and evacuate to $10^{-3}$ pa. Sealed quartz tube heated to the crystal melting temperature and

maintain for 30 min. The quartz tube rapidly cooled to room temperature to obtain colorless solids.

### Measurements

Gel filtration chromatography was performed using a SunFireTM C18 column conjugated to an ACQUITY UPLCH-class water HPLC system. Before running, each sample was purified via a 0.22 μm filter to remove any aggregates. The flow rate was fixed at 1.0 mL min⁻¹, the injection volume was 1 μL, and each sample was run for 10 min. UV-visible absorption spectra were obtained using Shimadzu UV-1750. Steady-state fluorescence/phosphorescence spectra and excitation spectra were measured using Hitachi F-4600. The lifetime and time-resolved emission spectra were obtained on Edinburgh FLSP980 fluorescence spectrophotometer equipped with a xenon arc lamp (Xe900), a nanosecond hydrogen flash-lamp (nF920), a microsecond flash-lamp (μF900), respectively. X-ray crystallography was achieved using a Bruker SMART APEX-II CCD diffractometer with graphite mono-chromated Mo-Kα radiation. Samples were analyzed using X-ray energy of 11 keV and incident angles ranging from $\Omega$ = 5 to 60 in 0.02 increments. Photoluminescence efficiency was collected on a Hama-matsu Absolute PL Quantum Yield Spectrometer C11347 (Supplementary Figs. 23–30). The luminescent photos and videos were taken by a Canon EOS 700D camera at room temperature.

### Computational details

The quantum mechanics/molecular mechanics (QM/MM) model with two-layer ONIOM method was employed to simulate the geometric and electronic structures of these host–guest systems by using Gaussian 09 program package[31], one PTZ plus two host molecules were chosen as the active QM region, and the remains were defined as fixed MM layer (Supplementary Fig. 15). The cluster models (4 × 4 × 4 supercell) were extracted from the host crystals. The electronic embedding is adopted in QM/MM calculations by incorporating the partial charges of the MM region into the quantum mechanical Hamiltonian. Based on the optimized ground state

($S_0$), the excitation energies were evaluated at the level of TD-B3LYP-D3(BJ)/def2-SVP. The spin–orbit coupling matrix elements were carried out based on the first-order Douglas–Kroll–Hess-like spin–orbit operator derived from the exact two-component (X2C) Hamiltonian[32] by using the Beijing Density Function (BDF) program[33–35]. It is noted that the adiabatic excitation energy was used for $T_1$ states. The hyperconjugation interactions were evaluated by natural bond orbital (NBO) analysis by using NBO version 3.1[36] incorporated in Gaussian 09. Based on the single point calculation for the QM molecules at the B3LYP-D3(BJ)/def2-SVP level, the reduced density gradient method[29] and energy decomposition analysis (EDA) with AMBER force field[37] approach were performed to evaluate the intermolecular interactions ($E_{int}$) by using Multiwfn package[38], the corresponding energies can be decomposed into electrostatic ($E_{ele}$), repulsion ($E_{rep}$), and dispersion ($E_{disp}$) energies.

In this work, energy decomposition analysis (EDA) with AMBER force field[37] method was performed by using Multiwfn package[38]. The total intermolecular interaction energy ($E_{int}$) can be decomposed into electrostatic ($E_{ele}$), repulsion ($E_{rep}$), and dispersion ($E_{disp}$) energies, i.e., $E_{int} = E_{ele} + E_{rep} + E_{disp}$. Where $E_{ele} = Q_A Q_B / R_{AB}$ can be described by Coulomb's law (Q represents the atomic charge and $R_{AB}$ is the interatomic distance). $E_{rep}$ is expressed by the following equation:

$$E_{rep} = \varepsilon_{AB} \left( \frac{R^0_{AB}}{r_{AB}} \right)^{12} \tag{1}$$

where $\varepsilon_{AB}$ is the depth of the van der Waals interaction potential well and $R^0_{AB}$ is the non-bonding contact distance, which is defined in AMBER force field. $E^{disp}$ can be deduced from the following equation:

$$E^{disp} = -2\varepsilon_{AB} \left( \frac{R^0_{AB}}{r_{AB}} \right)^6 \tag{2}$$

**Reporting summary**

Further information on research design is available in the Nature Portfolio Reporting Summary linked to this article.

## Data availability

All relevant data are included in this article and its Supplementary Information files. Source data are provided with this paper.

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

## Acknowledgements

This study was supported by the National Natural Science Foundation of China (62288102 (W.H.) and 21973043 (H.M.)), National Key R&D Program of China (grant no. 2020YFA0709900 (W.H.)), and the Hefei National Research Center for Physical Sciences at the Microscale (KF2020103 (H.M.)). We gratefully acknowledge HZWTECH for providing computational facilities.

## Author contributions

H.M., Z.A., and W.H. conceived the projects. X.J., K.L., and Q.M. made the calculations. H.S. helped the measurements of lifetime, and characterization of HPLC, XRD, and quantum yields. L.F. and X.Y. were primarily responsible for the experiments. H.M., Z.A., and W.H. analyzed the data and wrote the paper. All authors commented on the manuscript.

## Competing interests

The authors declare no competing interests.
