## [Peer Review File · Nature Communications]

Boosting organic phosphorescence in adaptive host-guest materials by hyperconjugationReviewer #1 (Remarks to the Author):

Phosphorescence under ambient condition from pure organic compounds is typically very faint and challenging to observe. It is surprisingly and stirring for the ultrahigh RTP efficiency in the adaptive host-guest systems that manipulated by the hyperconjugation and steric effects. Experimental and theoretical strongly demonstrated that the RTP efficiency closely correlates with the bending angle, establishing a robust, reliable, and effective strategy for designing adaptive and multifunctional organic RTP materials for optoelectronic applications. This is an original finding, which deserves publication in Nature Communications after some minor issues to be addressed.

Comment #1: The authors have designed a series of adaptive host-guest materials exhibiting high RTP efficiency, primarily attributed to the conformation match. As well known, the heavy-atom effect as a key factor influencing phosphorescence efficiency, why about the contribution from the S element?

Comment #2: This work revealed a significant correlation between RTP efficiency and blending angles, how about the role of structural deformation on the ISC efficiency?

Comment #3: In this manuscript, the authors have demonstrated that the structural blend governs the RTP efficiency of the adaptive host-guest systems. It is intriguing to explore the potential of forming host-dependent blending angles. Please elaborate

Reviewer #2 (Remarks to the Author):

Comments to the Author

The manuscript (NCOMMS-23-59890-T) contributed by Wei Huang et al. reports a series of host-guest materials based on Thianthrene (TA), phenoxathiine (PXT), xanthene (XT), phenothiazine (PTZ), and thioxanthene (TX) exhibiting room temperature phosphorescence that is tunable, depending on the doping that could reach a remarkable high phosphorescence quantum yield. The author extends this strategy to a ternary host-guest system, obtaining high phosphorescence efficiency and a time- and excitation- dependent phosphorescence. The authors perform photophysical, computational and crystallographic study of the system to rationalize the photophysical behaviors of the several host-guest crystalline systems they develop. I think this paper can recommended for publication on Nature communications after major revision and answering at the following points:

1. I would recommend thee author to remove the term spin-flip with intersystem crossing.
2. In supplementary Figure 6B, could the author explain why the phosphorescence spectra of PTZ at 77K is almost zero (orange dashed line).
3. In the experimental section, on the description of the method for the quantum yield determination, the Figure 7 could lead to misunderstanding. As drawn at the moment looks that A is the integral of the emission band at 300-400 nm. I advice the author to improve the clarity of the figure.
4. Supplementary figure 8 has a significant noise, could this affect the determination of the QE? Could the author introduce a description on the equations and setup to estimate the photoluminescence quantum yield.
5. The authors should describe the condition of the measurements reported in the figures. For example, in Supplementary Figure 15 is not reported the condition of the sample (solid, solution, 77K exc...)
6. In supplementary figure 19-21, some emission decays are clearly biexponential, could the authors comment these data.
7. The authors should add the shape of the excitation (IRF, instrument response file) when reporting emission lifetime decays (IRF).
8. The authors should estimate the non-radiative deactivation constant (k_{nr}) and radiative deactivation constant (k_r) of the long-lived excited states and correlate them to the trend observed by different doping. The results could be interesting and a strong addition to improve the quality of the manuscript.
9. Do the authors estimate the effect of oxygen on these host-guest materials? Since the long

emission lifetime of the materials, oxygen could greatly quench the phosphorescence. If any porosity or oxygen permeability is present this could decrease the recorded QE. I advise the author to estimate the QE also in deaerated condition.

Reviewer #3 (Remarks to the Author):

In their manuscript entitled "Boosting organic phosphorescence from adaptive host-guest materials by hyperconjugation", Ma et al aim at studying dibenzo-heterocyclic derivatives exhibiting a boost in their phosphorescence efficiency. It relies on a host-guest combination where the guest can adapt its structure to perfectly match the cavity of the host, using a particular hyperconjugation effect, namely $n \rightarrow \pi^*$ interactions. Both experimental and theoretical results are presented and discussed. For the experimental part, photophysical properties have been investigated in dilute solutions (at ambient temperature and 77 K) and within the solid state. Crystalline host-guest systems show higher phosphorescence efficiencies than pure crystals. Then, a mechanism for the efficient emission process is proposed, based on X-ray diffraction analysis and QM/MM calculations. The authors confirm the intuition that host-guest systems globally behave as the host. A general rule is thus suggested, making a link between the dihedral angle of the host and guest molecules and the resulting phosphorescence efficiency of the host-guest system. Computational approaches are then used to rationalize the boost of the emission process. Once again, a careful analysis of the dihedral angle has been done, along with a detailed analysis of different components of the interaction energy, highlighting the main contributions (either repulsive, dispersion, or electrostatic contribution). The computational part ends with the presentation of the energy diagrams and spin-orbit coupling matrix elements for different host-guest systems. The general rules proposed in the previous parts are tested by building new host-guest systems and are effectively confirmed. If the manuscript seems to be coherent there are still some comments and questions that should be addressed by the authors:

- Concerning the photophysical properties in dilute solution, why are the solvent different for absorption (THF) and emission (CH_2Cl_2)? Is there a particular reason?
- Why is the PTZ/TX system not reported in the first parts of the manuscript (photophysical properties) and just mentioned at almost the end of the "extension of the adaptive...?"
- Is there a reason to only consider PTZ as a guest and the others as hosts?
- How was the choice of the QM/MM model made? Would have it been relevant to consider a larger host environment for the guest molecule, to have a complete or at least larger surrounding for the guest molecule?
- Is the QM/MM model suitable to reproduce qualitatively the experimental optical properties? Did the authors check that?
- Are all the molecules in the QM region (Host+Guest) allowed to move during the optimization step? If the dihedral angles are not modified, what about the intermolecular distances?
- Can the change in emission colors be explained with the QM/MM calculations? Is it due to the guest or to the modification of the crystalline structure of the host?
- Concerning the explanation of the mechanism of RTP emission and particularly the XRD spectra, the authors have written "it is easily found that the host-guest materials have similar diffraction peaks with those of the host crystals". Maybe there are mistakes in the attribution or interpretation of the different spectra on Figure S22. The PTZ/XT spectrum is more like the PTZ one than the XT one. The PTZ/PXT may look more similar to PXT, but it is not obvious. It seems to be ok for PTZ/TA. Can the authors clarify this point?
- Fig. 4. I do not know what system is considered within the 4th spectrum of Fig 4a or on Fig 4.c. Is it PTZ/TX? PTZ/XT? The authors should clarify this point within the figure, the caption and maybe the text.
- Some typos are still present within the manuscript. "...five typical prototype molecules that were consisted of..", "...by using Gaussian 09 program package, Oone PTZ...". A careful proofreading should be done.

Point-by-Point Response to Referees**Reviewer #1:**

Phosphorescence under ambient condition from pure organic compounds is typically very faint and challenging to observe. It is surprisingly and stirring for the ultrahigh RTP efficiency in the adaptive host-guest systems that manipulated by the hyperconjugation and steric effects. Experimental and theoretical strongly demonstrated that the RTP efficiency closely correlates with the bending angle, establishing a robust, reliable, and effective strategy for designing adaptive and multifunctional organic RTP materials for optoelectronic applications. This is an original finding, which deserves publication in *Nature Communications* after some minor issues to be addressed.

Response: We appreciate the positive comments from the reviewer.

Comment #1: The authors have designed a series of adaptive host-guest materials exhibiting high RTP efficiency, primarily attributed to the conformation match. As well known, the heavy-atom effect as a key factor influencing phosphorescence efficiency, why about the contribution from the S element?

Response: We truly understand the reviewer's concern. When focusing on the PXT and PTZ guests, it is found that the steady-state photoluminescence (PL) and phosphorescence spectra are totally overlapped in dilute solution at 77 K, along with high PL efficiency (96.1% for PXT and 78.0% for PTZ), indicating the intersystem crossing process from singlet to triplet states is highly efficient, thus the population of the triplet excitons is insensitive to the heavy atom effect (HAE). This is why the conformation match can be regarded as the main contribution to the PLQY improvement.

Nevertheless, we further probed the contribution from the HAE following the reviewer's suggestion. It is worth noting that TA molecule contains two S atoms. For PXT molecule, there is only one S atom. No S atom is found in XT host. Thus, **the PTZ/TA has a stronger external HAE than that of PTZ/PXT and PTZ/XT in principle. However, the RTP efficiency is 69.0% for PTZ/TA, which is smaller than 96.8% for PTZ/PXT and 78.6% for PTZ/XT. Obviously, the external HAE cannot reveal the order of RTP efficiency.** Remarkably, it agrees well with the variation of conformation match $\Delta\theta_{HG}$, ranging from 27.7° in PTZ/TA, 10.1° in PTZ/PXT to 18.5° in PTZ/XT.

Supporting Figure I. (a) Chemical structures of TA, XT, PXT and PTZ molecules.

The numbers of S atoms in these molecules are listed. (b) Phosphorescence efficiency versus conformation match ($\Delta\theta_{\text{HG}}$). The $\Delta\theta_{\text{HG}}$ represents the difference in bending angle θ between host and guest molecules.

We then evaluated the contribution from the heavy atom effect (HAE) and conformation match to the PLQY improvement, by taking PXT/XT and PTZ/XT as examples. The RTP efficiency is increased from 54.9% in PXT/XT to 78.6% in PTZ/XT. Such an enhancement comes from two parts. One is that the internal and external HAEs promote the population of triplet excitons. The other is the conformation match suppresses the nonradiative transition of triplet states. Firstly, the external HAE acting on the guests in these two doping systems is similar owing to the same host TX. The internal HAE in PXT and PTZ guests is also similar, because PXT and PTZ have a similar constituent and contain one S atom. Additionally, their steady-state PL and phosphorescence spectra are almost overlapped in dilute solution at 77 K, indicating the intersystem crossing efficiency is nearly 100%, thus the population of the triplet excitons is insensitive to the external HAE. Secondly, when inspecting the conformation match $\Delta\theta_{\text{HG}}$, it is found that the $\Delta\theta_{\text{HG}}$ is decreased from 28.6° in PXT/XT to 18.5° in PTZ/XT, indicating the nonradiative decay is largely suppressed for the PTZ/XT with better conformation match. Therefore, we concluded that **the conformation match provides the main contribution to the increase of RTP efficiency (from 54.9% to 78.6%)**.

In brief, we strongly demonstrated that the improved PLQY in these adaptive host-guest materials is primarily contributed by the conformation match rather than the HAE (Supporting Figure 1).

Comment #2: This work revealed a significant correlation between RTP efficiency and blending angles, how about the role of structural deformation on the ISC efficiency?

Response: The ISC efficiency from singlet to triplet excited states strongly depends on the spin-orbit coupling (SOC) matrix elements. Taking PTZ-doped systems as an illustration, from PTZ/TA, PTZ/PXT to PTZ/XT, the PTZ guest molecule shows a host-dependent dihedral angle θ_{PTZ} , ranging from 137.8° , 147.2° to 177.8° . Within the context of increasing blending angles, the largest SOC value of PTZ guest is increased from 3.69 cm^{-1} for PTZ/TA to 4.38 cm^{-1} for PTZ/PXT, but then it goes down to 1.78 cm^{-1} for PTZ/XT, see Figure 3. This trend, correlated with bending angles, has been strongly demonstrated in other dibenzo-heterocyclic derivatives (*J. Phys. Chem. Lett.* 2021, 13, 1563; *Chem. Sci.* 2022, 13, 789; *Chem. Sci.* 2023, 14, 9733). Consequently, the ISC efficiency of PTZ guest in TA and PXT hosts is higher than that in XT host, ranging from 98.9% in PTZ/TA, 99.2% in PTZ/PXT to 96.5% in PTZ/XT (see Supplementary Table 10). This point can be further supported by the observation that the steady-state photoluminescence (PL) and phosphorescence spectra totally overlap for PTZ/TA and PTZ/PXT, while a reduction is occurrence for PTZ/XT, see Figure 2a. However, the contribution from structural deformation to the ISC efficiency can be considered negligible, given the high ISC efficiency of PTZ-doping systems exceeding 96%.

Comment #3: In this manuscript, the authors have demonstrated that the structural blend governs the RTP efficiency of the adaptive host-guest systems. It is intriguing to explore the potential of forming host-dependent blending angles. Please elaborate

Response: We appreciate the valuable comments from the reviewer. To evaluate the possibility of structural deformation, we then calculate the conformational reorganization energies, which are varied from 0.0498 eV for PTZ/TA, 0.0210 eV for PTZ/PXT, to 0.0926 eV for PTZ/XT. These small values showed that the guest PTZ can readily undergo a large structural deformation in response to the molecular environments of the host matrices, forming a specific geometry with bending angles ranging from 137.8° for PTZ/TA, 147.2° for PTZ/PXT to 177.8° for PTZ/XT.

Reviewer #2:

The manuscript (NCOMMS-23-59890-T) contributed by Wei Huang et al. reports a series of host-guest materials based on Thianthrene (TA), phenoxathiine (PXT), xanthene (XT), phenothiazine (PTZ), and thioxanthene (TX) exhibiting room temperature phosphorescence that is tunable, depending on the doping that could reach a remarkable high phosphorescence quantum yield. The author extends this strategy to a ternary host-guest system, obtaining high phosphorescence efficiency and a time- and excitation- dependent phosphorescence. The authors perform photophysical, computational and crystallographic study of the system to rationalize the photophysical behaviors of the several host-guest crystalline systems they develop. I think this paper can be recommended for publication on Nature communications after major revision and answering at the following points:

Response: We appreciate the positive comments from the reviewer.

Comment #1: I would recommend the author to remove the term spin-flip with intersystem crossing.

Response: We appreciate the reviewer for bringing it to our attention. We made the necessary adjustments in the revised manuscript.

Comment #2: In supplementary Figure 6B, could the author explain why the phosphorescence spectra of PTZ at 77K is almost zero (orange dashed line).

Response: In supplementary Figure 6B, the **orange solid line** represents the phosphorescence spectrum of PTZ crystal under ambient conditions. However, there is no signal of phosphorescence during the measurement. The **orange dashed line** refers to the phosphorescence spectrum of PTZ at 77 K. In order not to confuse the reader, we have deleted it.

Comment #3: In the experimental section, on the description of the method for the quantum yield determination, the Figure 7 could lead to misunderstanding. As drawn at the moment looks that A is the integral of the emission band at 300-400 nm. I advise the author to improve the clarity of the figure.

Response: Thanks for the valuable suggestion of the reviewer. The phosphorescence quantum yields of the compounds were redefined by the following equation:

$$\phi_{\text{phos}} = \frac{B}{A+B} \times \phi_{\text{PL}}$$

where A and B represent the integral areas of fluorescence and phosphorescence spectra, respectively. The phosphorescence was separated from the total PL spectrum based on the phosphorescence spectrum (Supplementary Figure 7). We have modified the related statement in the revision.

Supplementary Figure 7. Schematic diagram of integral areas of fluorescence and phosphorescence for quantum yield calculation.

Comment #4: Supplementary figure 8 has a significant noise, could this affect the determination of the QE? Could the author introduce a description on the equations and setup to estimate the photoluminescence quantum yield.

Response: We sincerely appreciate the reviewer for the careful review. We have rechecked the quantum efficiency and found that the quantum efficiency has a tiny change. We added the new version in the revised manuscript.

Supplementary Figure 8. Original data of the PTZ/TA phosphor for phosphorescence efficiency calculation.

Comment #5: The authors should describe the condition of the measurements reported in the figures. For example, in Supplementary Figure 15 is not reported the condition of the sample (solid, solution, 77K exc...)

Response: We sincerely appreciate the friendly suggestion. The test details have been added in the revised manuscript.

Comment #6: In supplementary figure 19-21, some emission decays are clearly biexponential, could the authors comment these data.

Response: We truly understand the concern of the reviewer. As seen from the **Supplementary Figures 19-21** and **Tables 2-4**, the TA, PXT and XT crystals exhibit a clearly biexponential decay for the phosphorescence lifetime, along with a low phosphorescence efficiency of less than 3.5%, which may be ascribed to the nonradiative decay channel caused by the intermolecular aggregations. From **Supporting Figure 2**, there are multiple intermolecular stacking patterns.

For the PTZ-doping systems, the phosphorescence lifetime decay profiles seem to closely correlated to the phosphorescence efficiency. For example, the phosphorescence of PTZ/PXT systems exhibits high emission efficiency ($> 88\%$), because of the good conformation matching between host and guest molecules, which is also favorable for forming a uniform conformation for the isolated guest in the host-guest aggregates, this thus is responsible for the single exponential decay of phosphorescence. In contrast, the decrease of the phosphorescence efficiency ($< 69\%$) for the PTZ/TA systems can be ascribed to the reduced conformation matching, which may produce diverse conformations for the isolated guest, thereby leading to biexponential decay for the phosphorescence.

Supporting Figure 2. Intermolecular packings of the TA, PXT, and XT crystals.

Comment #7: The authors should add the shape of the excitation (IRF, instrument response file) when reporting emission lifetime decays (IRF).

Response: We sincerely appreciate the valuable comments from the reviewer. We have added these data in the revised manuscript.

Comment #8: The authors should estimate the non-radiative deactivation constant (k_{nr}) and radiative deactivation constant (k_r) of the long-lived excited states and correlate them to the trend observed by different doping. The results could be interesting and a strong addition to improve the quality of the manuscript.

Response: Thanks for the valuable comments from the reviewer. We evaluated the decay rates of the lowest triplet excitons, as listed in Supplementary Table 10. It is found that all doped systems show a high ISC efficiency (Φ_{isc}) of larger than 96%, indicating the RTP efficiency is mainly governed by the decay process of T_1 state. As the increase of the RTP efficiency from 54.9% to 98.9% for these host-guest systems, the nonradiative decay rate (k_{nr}) is gradually decreased from 19.51 s^{-1} to 0.00 s^{-1} , while the radiative decay rate (k_p) shows a value of ca. 20 s^{-1} with a small change. Consequently, the increase of the RTP efficiency strongly depends on the suppression of the nonradiative decay rate, which stems from the promoted degree of the conformation match. We have added these data and discussion in the revised manuscript.

Supplementary Table 10. Photophysical parameters of PTZ/TA, PTZ/PXT, PTZ/XT, PXT/XT, PXT/TX and PTZ/TX under ambient conditions.

	Φ_{PL} (%)	Φ_P (%)	Φ_{isc}^a (%)	τ_p (ms)	k_p (s^{-1}) ^b	k_{nr} (s^{-1}) ^c
PXT/TX	100.0	98.9	98.9	47.91	20.87	0.00

PTZ/PXT	97.6	96.8	99.2	44.81	21.78	0.54
PTZ/TX	86.3	84.3	98.0	39.50	21.78	3.34
PTZ/XT	82.1	78.6	96.5	41.44	19.66	4.48
PTZ/TA	70.1	69.0	99.1	29.74	23.41	10.21
PXT/XT	57.1	54.9	97.8	22.48	24.97	19.51

$$^a)\Phi_{\text{isc}} = 1 - \Phi_{\text{F}} - \Phi_{\text{ic}} \approx 1 - \Phi_{\text{PL}} + \Phi_{\text{P}}; ^b)k_{\text{p}} = \Phi_{\text{P}}/(\Phi_{\text{isc}} \times \tau_{\text{p}}); ^c)k_{\text{nr}} = 1/\tau_{\text{p}} - k_{\text{p}}$$

Comment #9: Do the authors estimate the effect of oxygen on these host-guest materials? Since the long emission lifetime of the materials, oxygen could greatly quench the phosphorescence. If any porosity or oxygen permeability is present this could decrease the recorded QE. I advice the author to estimate the QE also in deaerated condition.

Response: As suggested, we conducted a series of contrasting experiments to probe the impact of oxygen on the RTP performance, see Supporting Figure 3. After exposing the sample to oxygen for a duration of 2 minutes, the emission intensity of these adaptive host-guest materials shows a tiny change compared to those under a nitrogen environment. Such a change suggests the close molecular packings in these host-guest materials, hindering the penetration of oxygen. Namely, a superior conformation match enhances the RTP efficiency of the adaptive host-guest materials by suppressing the nonradiative deactivations from the vibronic coupling and oxygen quenching.

Supporting Figure 3. The phosphorescence spectra of (a) PTZ/TA, (b) PTZ/PXT and (c) PTZ/XT under nitrogen and oxygen environments.

Reviewer #3:

In their manuscript entitled “Boosting organic phosphorescence from adaptive host-guest materials by hyperconjugation”, Ma et al aim at studying dibenzo-heterocyclic derivatives exhibiting a boost in their phosphorescence efficiency. It relies on a host-guest combination where the guest can adapt its structure to perfectly match the cavity of the host, using a particular hyperconjugation effect, namely $n \rightarrow \pi^*$ interactions.

Both experimental and theoretical results are presented and discussed. For the experimental part, photophysical properties have been investigated in dilute solutions (at ambient temperature and 77 K) and within the solid state. Crystalline host-guest systems show higher phosphorescence efficiencies than pure crystals. Then, a mechanism for the efficient emission process is proposed, based on X-ray diffraction analysis and QM/MM calculations. The authors confirm the intuition that host-guest systems globally behave as the host. A general rule is thus suggested, making a link between the dihedral angle of the host and guest molecules and the resulting phosphorescence efficiency of the host-guest system. Computational approaches are then used to rationalize the boost of the emission process. Once again, a careful analysis of the dihedral angle has been done, along with a detailed analysis of different components of the interaction energy, highlighting the main contributions (either repulsive, dispersion, or electrostatic contribution). The computational part ends with the presentation of the energy diagrams and spin-orbit coupling matrix elements for different host-guest systems. The general rules proposed in the previous parts are tested by building new host-guest systems and are effectively confirmed. If the manuscript seems to be coherent there are still some comments and questions that should be addressed by the authors:

Response: We truly appreciate the positive comments from the reviewer.

Comment #1: Concerning the photophysical properties in dilute solution, why are the solvent different for absorption (THF) and emission (CH_2Cl_2)? Is there a particular reason?

Response: We sincerely appreciate the reviewer for the careful review. Indeed, both the UV absorption and steady-state PL spectra of these compounds (PTZ, TA, PXT, TX, and XT) were measured in the dilute 2-mTHF solution (2×10^{-5} M) under ambient conditions (Supplementary Figure 3). We sincerely apologize for any confusion caused by our oversight and have corrected this error in the revised manuscript.

Comment #2: Why is the PTZ/TX system not reported in the first parts of the manuscript (photophysical properties) and just mentioned at almost the end of the “extension of the adaptive...”?

Response: We truly understand the concern of the reviewer. From the PTZ-doped systems as depicted in Figure 2, we strongly demonstrated that the host-dependent conformation match governs the RTP efficiency of these adaptive host-guest materials, which was further strengthened by the PXT-doped systems in Figure 4. Subsequently, to probe the guest-dependent RTP property, we then constructed PTZ/TX as shown in Figure 4. In contrast to the high RTP efficiency of 98.9% in PXT/TX, the RTP efficiency

of PTZ/TX decreases to 84.3%. Such a change can be also attributed to the reduced conformation match, evidenced by an increased difference in bending angles between host and guest, ranging from 5.7° in PXT/TX to 15.8° in PTZ/TX. Therefore, we concluded that **the conformation match provides the main contribution to the increase of RTP efficiency**. This point is strongly supported by all the PTZ- and PXT-doping systems in our work, the RTP efficiency is gradually increased from PXT/XT (54.9%), PTZ/TA (69.0%), PTZ/XT (78.6%), PTZ/PXT (96.8%) to PXT/TX (98.9%), which is well consistent with the change in conformation match $\Delta\theta_{\text{HG}}$, decreasing from 28.6° in PXT/XT, 27.7° in PTZ/TA, 18.5° in PTZ/XT, 10.1° in PTZ/PXT to 5.7° in PXT/TX (see Figure 4e and Supporting Figure 4).

In addition, we prepared a ternary host-guest system by embedding PTZ and PXT guests into the crystalline TX matrix. To elucidate the origin of this doping system, the reported PXT/TX and PTZ/TX in Figure 4 are convenient for comparing the RTP properties of these three doping systems. We hope that the reviewer can be acceptable.

Supporting Figure 4. (a) Chemical structures of TA, XT, PXT, PTZ and TX, and the numbers of S atoms in these molecules. (b) Phosphorescence efficiency versus conformation match ($\Delta\theta_{\text{HG}}$). The $\Delta\theta_{\text{HG}}$ represents the difference in bending angle θ between host and guest molecules.

Comment #3: Is there a reason to only consider PTZ as a guest and the others as hosts?

Response: For host-guest system with bright luminescence, the host should have a higher energy level than that of guest to hinder the luminescence quenching through energy transfer channel from guest to host. By inspecting the phosphorescence spectra of these dibenzo-heterocyclic analogues in dilute 2-mTHF solution (2×10^{-5} M) at 77 K, the energy levels of T_1 states show an ascending order: PTZ < TA < PXT < TX < XT (see Supplementary Fig. 5). Thus, the PTZ can only serve as the guest due to the lowest energy level, and when PXT acts as a guest, only TX and XT can serve as the host. We hope that the reviewer can be acceptable.

Supplementary Figure 5. Normalized steady-state PL spectra (black dashed line) and phosphorescence spectra (green solid line) of XT, TX, PXT, TA, and PTZ in dilute 2-mTHF solution (2×10^{-5} M) under 77 K excited at 310 nm.

Comment #4: How was the choice of the QM/MM model made? Would have it been relevant to consider a larger host environment for the guest molecule, to have a complete or at least larger surrounding for the guest molecule?

Response: The computational models were constructed by cutting a large cluster ($4 \times 4 \times 4$ supercells) from the host crystal (see Supplementary Figure 24). Taking PTZ/PXT as an example, this cluster shows a cuboid form with a size of $30 \text{ \AA} \times 30 \text{ \AA} \times 80 \text{ \AA}$. To assess the influence of host environment on the electronic structures of the QM region, we then reduced this cluster to the sizes of $30 \text{ \AA} \times 30 \text{ \AA} \times 60 \text{ \AA}$ and $30 \text{ \AA} \times 30 \text{ \AA} \times 40 \text{ \AA}$, named PTZ/PXT-1 and PTZ/PXT-2, respectively. It is found that the excitation energies of T_1 state remain unchanged with a value of 2.48 eV for both PTZ/PXT-1 and PTZ/PXT-2, compared to 2.48 eV for PTZ/PXT. We added these computational details in the revised manuscript and hope the reviewer finds them acceptable.

Comment #5: Is the QM/MM model suitable to reproduce qualitatively the experimental optical properties? Did the authors check that?

Response: We truly understand the concern from the reviewer. In our QM/MM model, the electrostatic embedding potential was used to describe local excitations of molecules in crystal environments. The electrostatic interaction potential from the MM part on QM can be described by the Coulomb potential

$$V_{emb}^{QM/MM}(\vec{r}) = \sum_{j \in MM} \frac{q_j}{|\vec{r} - \vec{R}_j|} = \frac{\sum_{j \in MM} q_j}{|\vec{r}|} + \frac{\vec{r} \cdot \sum_{j \in MM} q_j \vec{R}_j}{|\vec{r}|^3} + \dots$$

where q_j is the point charges of one atom in MM part, R_j is the corresponding coordinate of atom, \vec{r} is the coordinate of electron in QM part. The first term is the contribution of atomic net charge, and the second one is the dipole contribution term. Indeed, this QM/MM model had been widely employed to evaluate the fluorescence and phosphorescence emissions of organic crystals, giving a good consistency between experiments and calculations (*Acc. Chem. Res.*, 2021, 54, 940; *Natl. Sci. Rev.*, 2017, 4, 224).

In our work, the RTP peak in the experiment is gradually redshifted from 498 nm (2.49 eV) for PTZ/TA, 508 nm (2.44 eV) for PTZ/PXT to 517 nm (2.40 eV) for PTZ/XT. This change is well reproduced by theoretical calculations, where the T_1 energies are decreased from 2.51 eV for PTZ/TA, 2.48 eV for PTZ/PXT to 2.31 eV for PTZ/XT. Note that the calculated excitation energies of T_1 states agree well with the experimental values, and the absolute deviation is less than 0.10 eV, which is reasonable for the time-dependent density functional theory with a typical error of 0.2-0.4 eV (*Int. J. Quantum Chem.* 2013, 113, 2019; *Chem. Rev.* 2021, 121, 9873).

Comment #6: Are all the molecules in the QM region (Host+Guest) allowed to move during the optimization step? If the dihedral angles are not modified, what about the intermolecular distances?

Response: We appreciate the careful review of this section. For the QM/MM calculations, all the molecules in the QM region can move to study the intermolecular interaction between host and guest. We have demonstrated that the change in bending angle is small ($< 4.5^\circ$) for the host molecules neighboring the PTZ guest (Supplementary Table 6), while the intermolecular distances of hosts are also insensitive to the doping PTZ guest (Supporting Figure 5). For example, the shortest distance between centroids of the host molecules is changed from 5.910 Å in TA host single crystal to 5.913 Å in PTZ/TA systems. Thus, it rationally speculated that the host crystal structure is intact after insertion of the guest molecule.

Supporting Figure 5. The distance between centroids of the host molecules in host single crystals (top) and PTZ-doped systems (bottom). The unit is angstrom.

Comment #7: Can the change in emission colors be explained with the QM/MM calculations? Is it due to the guest or to the modification of the crystalline structure of the host?

Response: Thanks to the careful review from the reviewer. In the experiment, the RTP peak is gradually redshifted from 498 nm (2.49 eV) for PTZ/TA, 508 nm (2.44 eV) for PTZ/PXT to 517 nm (2.40 eV) for PTZ/XT. This change is well reproduced by theoretical calculations, where the T_1 energies are decreased from 2.51 eV for PTZ/TA, 2.48 eV for PTZ/PXT to 2.31 eV for PTZ/XT. Note that the calculated excitation energies of T_1 states agree well with the experimental values, and the absolute deviation is less than 0.10 eV, which is reasonable for the time-dependent density functional theory with a typical error of 0.2-0.4 eV (*Int. J. Quantum Chem.* 2013, 113, 2019; *Chem. Rev.* 2021, 121, 9873).

To gain the origin of such a change, we then calculated the excitation energies of T_1 state for PTZ guest without considering the crystal environment. It is found that the T_1 energies of PTZ are decreased from 2.48 eV, 2.48 eV to 2.33 eV, as the increase of the bending angle from 137.8° for PTZ/TA, 147.2° for PTZ/PXT to 177.8° for PTZ/XT. Namely, the energy level of PTZ guest is mainly contributed by the structural deformation rather than the host environment, since the tiny change (less than 0.03 eV) in comparison to the excitation energy of PTZ guest with and without host environment. This point has been demonstrated in other dibenzo-heterocyclic derivatives (*Chem. Sci.* 2022, 13, 789; *J. Phys. Chem. Lett.* 2021, 13, 1563).

Comment #8: Concerning the explanation of the mechanism of RTP emission and particularly the XRD spectra, the authors have written “it is easily found that the host-guest materials have similar diffraction peaks with those of the host crystals”. Maybe there are mistakes in the attribution or interpretation of the different spectra on Figure S22. The PTZ/XT spectrum is more like the PTZ one than the XT one. The PTZ/PXT may look more similar to PXT, but it is not obvious. It seems to be ok for PTZ/TA. Can the authors clarify this point?

Response: Thanks to the careful review from the reviewer. We carefully rechecked our original samples and found that the PTZ/XT and PTZ/PXT materials had undergone some weathering, resulting in inconsistent XRD results between host-guest materials and host crystals. We then prepared fresh PTZ/XT and PTZ/PXT materials and remeasured their XRD spectra, it showed that these host-guest materials have similar diffraction peaks with those of the host crystals (see Supplementary Figure 22).

Supplementary Figure 22. Normalized XRD spectra for host single crystal and mixed crystals.

Comment #9: Fig. 4. I do not know what system is considered within the 4th spectrum of Fig 4a or on Fig 4.c. Is it PTZ/TX? PTZ/XT? The authors should clarify this point within the figure, the caption and maybe the text.

Response: PT/TX is a ternary host-guest system, which is consist of PTZ and PXT two guests and the crystalline TX host. We added this statement in the revised caption of Figure 4.

Comment #10: Some typos are still present within the manuscript. "...five typical prototype molecules that were consisted of...", "...by using Gaussian 09 program package, Oone PTZ...". A careful proofreading should be done.

Response: Thanks to the thorough review from the reviewer. We have carefully reviewed the manuscript and corrected these errors in the revised manuscript.

Reviewer #1 (Remarks to the Author):

The authors have nicely addressed the comments and the paper can be accepted.

Reviewer #2 (Remarks to the Author):

The authors of manuscript NCOMMS-23-59890-T respond affirmatively to all the referee's points. The manuscript, in its present state, is suitable for publication

Reviewer #3 (Remarks to the Author):

I truly appreciate the efforts made by the authors to answer my questions and address my concerns about their manuscript and do not have any further questions or comments.